# Youth's engagement and perceptions of disposable e-cigarettes: a UK focus group study

Marissa J Smith ,[1] Anne Marie MacKintosh,[2] Allison Ford,[2] Shona Hilton [1]

[1]MRC/CSO Social and Public Health Sciences Unit, University of Glasgow, Glasgow, UK
[2]Institute for Social Marketing, University of Stirling, Stirling, UK

**Correspondence to**
Dr Marissa J Smith;
marissa.smith@glasgow.ac.uk

## ABSTRACT

**Objectives** Evidence suggests that use of flavoured disposable electronic cigarettes (e-cigarettes) is increasing. Considering the growing popularity and rapid evolution of e-cigarettes, we explored youth's perceptions and engagement with disposable e-cigarettes.

**Design** Twenty focus groups were conducted between March and May 2022, with 82 youths aged 11–16 living in the Central belt of Scotland.

**Methods** Youths were asked about smoking and vaping behaviours and disposable e-cigarettes and were shown vaping-related images and videos from social media which were used to stimulate discussion about different messages, presentations and contextual features. Transcripts were imported into NVivo V.12, coded thematically, and analysed.

**Results** Youths described disposable e-cigarettes as 'cool', 'fashionable' and enticing and viewed as a modern lifestyle 'accessory'. Tank models were perceived as being used by older adults. Youths stated that disposable e-cigarettes were designed in a way to target youths and the brightly coloured devices and range of flavourings encouraged youths to want to try the products, particularly sweet flavourings. Participants perceived e-cigarettes to be less harmful compared with combustible cigarettes but noted the uncertainty of ingredients in disposable e-cigarettes.

**Conclusions** Youths distinguish between e-cigarettes with varying characteristics and social perceptions of users. These findings provide evidence that disposable e-cigarettes are attractive to youths. Future research is needed to understand the factors that contribute to youth perceptions of disposable e-cigarettes. Policymakers should work together to design and implement policies and strategies to prevent youth uptake of vaping.

## BACKGROUND

The use of e-cigarettes among youths in Great Britain (GB) has increased in 2022 compared with 2021; however, use among never-smokers remains low and mostly experimental.[1 2] Since the development of e-cigarettes, public health researchers and tobacco control advocates have debated the role of e-cigarettes as a harm reduction tool. Proponents of e-cigarette harm reduction believe e-cigarettes can play a role in eliminating smoking-related diseases and consider

## STRENGTHS AND LIMITATIONS OF THIS STUDY

⇒ This research offers timely insights into youths perceptions about the growing use of disposable e-cigarettes.
⇒ It provides an in-depth analysis from interviews with a diverse sample of 82 youths aged 11–16.
⇒ Our findings present new evidence on how youths experience targeted e-cigarette marketing via social media content as visual prompts.
⇒ Our qualitative thematic analysis of the data allows depth of opinions but cannot offer predictions about the frequency of specific opinions with a wider population.

them to be a breakthrough in harm reduction development.[3–5] Whereas opponents of the e-cigarette harm reduction debate argue that caution should be taken when endorsing e-cigarette products until crucial evidence becomes available.[6] E-cigarettes are often termed a short-term tobacco harm reduction tool, as they do not contain tobacco or tar which are known to cause numerous smoking-related diseases, including cardiovascular disease. A newly published Cochrane review[7] found that nicotine e-cigarettes were superior to placebo e-cigarettes and at least as effective as nicotine replacement therapy (NRT) for smoking cessation, which is consistent with findings from other randomised controlled trials.[8–10] In addition, the review stated that there is moderate certainty in the evidence that nicotine-containing e-cigarettes increase the quit rate compared with NRT and non-nicotine-containing e-cigarettes.[7] Despite differences in opinion within the public health community regarding the value of e-cigarettes in harm reduction for adults, there is broad consensus on the need to protect young people from initiating vaping.[11]

Since the development of the first e-cigarette in 2003, there are now a variety of models or 'generations' available. First-generation e-cigarettes (sometimes referred

to as 'cigalikes') were disposable and designed to mimic the look and feel of combustible cigarettes.[12] Over time, new e-cigarette types were developed to more effectively deliver nicotine contained in e-liquid. Second-generation e-cigarettes are larger and are generally refillable using e-liquids.[13] Third-generation e-cigarettes (tanks or mods) are much larger than the previous generations and are refillable and rechargeable.[12 14] They are modifiable devices ('mods'), meaning the user can customise the substances in the device[15] and adjust the power of the device to give a stronger throat hit.[16 17] The fourth generation of e-cigarettes is called 'Pod Mod'. They contain a prefilled or refillable 'pod' or pod cartridge with a modifiable 'mod' system ('Pod-Mod').[14]

Recently, disposable e-cigarettes (such as 'Puff-bar', 'Elf-bar' or 'Geek-bar') have started to dominate the market.[18] Disposable e-cigarettes retail for around £5–£7 (US$7–US$9) in the UK—about half the price of a pack of 20 cigarettes.[19] In GB, data captured in 2022, found that disposable e-cigarettes have become the most common device type (52.0% compared with 7.7% in 2021), with Elf Bar and Geek Bar being the most popular brands.[1] Despite the popularity of disposable e-cigarettes, little is known about the design, chemical characteristics, or how they may impact health.

Considering the rapid growth and popularity of disposable e-cigarettes, this research aims to explore youth's perceptions and engagement with disposable e-cigarettes, awareness of product characteristics, appeal of products and flavours, perceptions of harm, perceived target group and purchasing behaviours. User-generated and influencer marketing content on social media represents a key influence on young people's understandings of products. It is essential to monitor the content that young people access related to e-cigarettes and through focus groups with youths, so we can understand how young people relate to that content, why e-cigarettes might appeal to youths and why they need protected, which would not be feasible with population surveys.

## METHODS

We conducted 20 focus groups between March and May 2022. Focus groups included between three and five participants (a total of 82 participants). Purposive sampling was used to recruit a diverse sample of youths in terms of sex, socioeconomic background and smoking and vaping status. Eleven groups were recruited through youth workers in local youth organisations. These gatekeepers handed out information sheets and helped achieve the sampling frame in terms of youth demographics and experiences with regard to traditional cigarettes and e-cigarettes. The three organisations that helped with participant recruitment worked specifically with young people from disadvantaged backgrounds in urban areas. This recruitment strategy resulted in the inclusion of a range of participants from more affluent and more deprived backgrounds and with experiences of smoking and vaping. Seven groups were recruited through the Schools Health and Wellbeing Improvement Research Network (SHINE) Newsletter which is distributed monthly to over 500 schools in Scotland. The remaining two groups were recruited via personal networks directly by MS.

Focus group discussions were facilitated to allow the research team to explore how opinions about disposable e-cigarettes are developed. Friendship groups of 3–5 participants were used to facilitate in-depth insights and promote participant interaction. Each participant was given a £20 shopping voucher as compensation for their time.

Prior to the start of the focus groups, participants completed a short anonymous questionnaire about their age, sex, postcode, smoking and e-cigarette use status. For both traditional cigarettes and e-cigarettes, the questionnaire asked participants to specify whether they had tried or used them in the past or were using them at the time of the study. Based on a review of the literature a topic guide was developed which covered three key areas, including different types of e-cigarette products and flavours, perceptions of harm and purchasing behaviours.

Images of different types of e-cigarettes ('tanks', disposables and pod devices) and e-liquids were used as conversation starters. Group discussions were facilitated by MJS. Ten of the groups were conducted online using Microsoft Teams and 10 were conducted face-to-face. Of these, one of the groups was conducted on the youth organisation's premises, and the other nine were conducted at the school, with representatives of the youth organisation present. Groups lasted between 40 and 66 min. Field-notes reflecting on the focus group and individual issues discussed were written up for each group. All focus groups were audio-recorded with participants' permission and transcribed verbatim. We conducted thematic analysis of the data from the interview transcripts and discussion group minutes. The process followed Braun and Clarke's[20] six-phase framework for thematic analysis. The steps involved were: (1) familiarisation with the data; (2) generating initial codes; (3) searching for themes; (4) reviewing themes; (5) defining and naming themes; and (6) writing the report.[20] The research team read and reread the transcripts to become familiar with the data, and then iteratively constructed a coding frame based on the topic to enable consistent organisation of relevant data. NVivo was used to organise categories on the basis of inductive themes that emerged from close reading of the, capture of both areas of agreement and less typical perspectives across a range of categories. Each transcript was imported into NVivo V.12, coded independently, cross-checked and analysed by MJS and SH. Contradictory cases and group dynamics were discussed, making use of transcripts and field notes. The researcher reflected on her role as researcher, remained constantly aware of her position and took care not to introduce bias throughout the research. To further reduce bias the researcher recorded the focus groups and analysed them some time after they were completed ensuring a more

reflective view point of occurrences. Ethical approval for the study was obtained from the University of Glasgow's Medical and Veterinary Life Sciences Ethics Committee (reference 200210034).

## Patient and public involvement
None.

## RESULTS
### Participant characteristics
Eighty-two youths aged 11–16 years participated (47 females (57%) and 35 males (43%)) in this study. This sample represented a wide diversity in sociodemographic characteristics and smoking-related behaviours. The age distribution within the sample was skewed slightly towards 14–15-year-olds, with 14-year-olds making up the largest subgroup (n=24). While the majority of participants did not currently smoke or use e-cigarettes, the sample included 10 smokers and 18 youths who used e-cigarettes. Table 1 describes the focus group composition and participants in more detail and table 2 summarises smoking and e-cigarette use among the sample.

### Product characteristics
Youths referred to disposable e-cigarettes as vapes or disposable vapes. Participants described products based on product characteristics including rechargeable/disposable and design (small vs large). Some reported that the disposable variety were not e-cigarettes and the rechargeable were.

> They [disposable e-cigarettes] aren't real ''cause they are disposable, they aren't real vapes. (Male, current smoker, current vaper)

Product characteristics such as design were also used to classify products. Participants discussed disposable

| Table 1 | Focus group location, participants and their cigarette smoking and e-cigarette use | | | | |
|---|---|---|---|---|---|
| Group | Area | Sex | Age | Cigarette smoker | E-cigarette use |
| 1 | Affluent | Female | 13–15 | Never | Never |
| 2 | Affluent | Female | 14–15 | Never | Mixed: never (4)/tried (1) |
| 3 | Affluent | Female | 13–16 | Never | Mixed: never (2)/tried (1) |
| 4 | Deprived | Mixed: male (3)/female (2) | 12–15 | Mixed: never (3)/current (2) | Mixed: never (3)/tried (1)/current (1) |
| 5 | Deprived | Mixed: male (1)/female (4) | 14–16 | Mixed: never (2)/tried (2)/current (1) | Mixed: never (2)/tried (2)/current (1) |
| 6 | Deprived | Male | 12–15 | Never | Never |
| 7 | Deprived | Male | 16 | Current | Current |
| 8 | Affluent | Mixed: male (2)/female (3) | 14 | Never | Never |
| 9 | Deprived | Male | 16 | Mixed: tried (1)/current (2) | Current |
| 10 | Deprived | Mixed: male (4)/female (1) | 14–15 | Mixed: never (3)/tried (1)/current (1) | Mixed: never (3)/tried (1)/current (1) |
| 11 | Deprived | Mixed: male (3)/female (2) | 13–16 | Mixed: never (2)/tried (2)/current (1) | Mixed: never (1)/current (4) |
| 12 | Affluent | Mixed: male (2)/female (1) | 15–16 | Tried | Mixed: tried (2)/current (1) |
| 13 | Affluent | Female | 13–16 | Never | Never |
| 14 | Deprived | Mixed: male (1)/female (3) | 11–12 | Never | Never |
| 15 | Deprived | Mixed: male (3)/female (1) | 11–12 | Never | Never |
| 16 | Deprived | Mixed: male (2)/female (2) | 11–12 | Never | Never |
| 17 | Deprived | Female | 14–16 | Mixed: never (4)/tried (1) | Mixed: never (1)/tried (1)/current (3) |
| 18 | Deprived | Male | 13–16 | Never | Never |
| 19 | Deprived | Female | 14 | Never | Mixed: tried (2)/current (1) |
| 20 | Affluent | Female | 15–16 | Never | Tried (3) |

**Table 2** E-Cigarette use according to cigarette smoking

| Cigarette smoker | E-cigarette use | | | | | | | | | | | |
| | Never | | | Tried | | | Current | | | Total | | |
| | n | (col %) | (row %) | n | (col %) | (row %) | n | (col %) | (row %) | n | (col %) | (row %) |
|---|---|---|---|---|---|---|---|---|---|---|---|---|
| Never | 49 | 98.0 | 79.0 | 9 | 64.3 | 14.5 | 4 | 22.2 | 6.5 | 62 | 75.6 | 100.0 |
| Tried | 1 | 2.0 | 10.0 | 4 | 28.6 | 40.0 | 5 | 27.8 | 50.0 | 10 | 12.2 | 100.0 |
| Current | 0 | 0.0 | 0.0 | 1 | 7.1 | 10.0 | 9 | 50.0 | 90.0 | 10 | 12.2 | 100.0 |
| Total | 50 | 100.0 | 89.0 | 14 | 100.0 | 14.6 | 18 | 100.0 | 22.0 | 82 | 100.0 | 100.0 |

e-cigarettes being small colourful products, whereas the rechargeable tank models were bulky.

> I think they've been designed differently, so you can tell which ones apart. Like, the electrical ones, the ones that you charge, they're like bigger, and a bit, like, bulkier. (Female, never smoker, never vaper)

Participant views diverged when shown illustrative examples of different types of vaping products, particularly disposable e-cigarettes. Several participants were able to easily recognise disposable e-cigarettes but not other types:

> There is definitely like one that I recognise like the small wee pink one with the black top. But I didn't recognise the rest to be honest. (Female, never smoker, never vaper)

Several participants were not able to identify disposable e-cigarettes when shown illustrative examples and often thought they were other products, such as highlighters or lighters.

> That's not a vape, it was a highlighter. (Male, never smoker, never vaper)

> When I first saw it, it looked like a lighter. (Male, never smoker, never vaper)

> Like a tin of mints or something. (Female, never smoker, tried vaping)

### Appeal of products

Participants described several positive attributes of disposable e-cigarettes including the design, as they were portable and discreet.

> If you're an underage child vaping you're not going to want to have that big bulky thing 'cause you might get caught with it. Something as small as the thin thing, that could easily fit in your pocket and not have anyone notice. But that thing [tank model], you'd have it sticking out to see. (Male, never smoker, never vaper)

This was also discussed by participants when referring to using the products at school.

> Yeah, they are much smaller so, they can hide them when at school." (Female, never smoker, current vaper)

### Appeal of flavours

Participants particularly liked the variety of flavours that are available such as apple and pink lemonade. Several participants discussed that the variety of flavours encouraged users to try other available flavours.

> You get like different flavours in sweets and stuff, you might like the taste of blueberry and because in the vape you've got that same taste, that's where it'd be like, oh I really like blueberry, I'd want to see if it is, and then that's what also gets you addicted to it. (Male, tried smoking, current vaper)

Interestingly, when participants discussed flavours, they specifically referred to disposable e-cigarettes, with several participants unaware that e-liquids were available in a variety of flavourings.

> Like the range of flavours, and how we were saying about how the disposable vapes had, like, a lot of different flavours. But we weren't aware of the flavours that came with e-liquid ones. (Female, never smoker, never vaper)

Participants associated the colour of disposable e-cigarettes with flavourings, for example, one dual user stated, '*certain flavours would have different designs. Strawberry would have pink or red*' (Male, current smoker, current vaper). While, one nonuser explained, '*the likes of strawberry, that would be red because strawberries are red. And they would do different colours like that, 'cause of the flavours*' (Female, never smoker, never vaper).

### Perceived negative attributes

Disposable e-cigarettes are designed for single use and the environmental impact of the waste was raised by participants.

> They [disposable e-cigarettes] are bad for the environment because people just throw them away. (Female, never smoker, current vaper)

Participants also spoke about the products being non-recyclable and that this affects the environment. One participant stated, '*I don't think they're recyclable, either, so it's like a lot more waste*' (Female, never smoker, never vaper), another participant added, '*they [disposable e-cigarettes] take longer to break down, definitely*' (Male, never smoker, never vaper).

One e-cigarette user explained that the environmental impact of using a disposable e-cigarette does not affect his choice to use them.

> I like to use the ones which are disposable and not ones which are refillable. It is a collective effort to save the environment, but I don't want to put extra money to save the environment. (Male, tried smoking, current vaper)

### Perceived target audience

The design of the products was further referred to by participants when discussing the target audience of the different types of vaping products. Participant views of users were dependent on the subtype of products used. For example, the larger tank models were perceived to be targeted at and used by users older in age, while disposable e-cigarettes were described as 'cool', 'trendy' and a 'fashion accessory' and were perceived to be targeted at and used by youths.

> The disposables are used by like all younger people like aged 15 and 16. But adults, they've got the bigger ones like the rechargeable ones. (Female, tried smoking, tried vaping)

> The disposable ones have got different colours, they're brighter, that's probably more aimed at younger people. Whereas, you know, like the big chunky ones are probably more aimed for people who have actually come off smoking. (Female, never smoker, tried vaping)

### Perceptions of harm

Many youths perceived disposable e-cigarettes as less harmful than combustible cigarettes.

> They're not as bad as actual cigarettes for you. So, it can cut down the amount of cigarettes that you smoke. (Female, current smoker, current vaper)

Although disposable e-cigarettes were perceived as less harmful compared with tobacco cigarettes, non-user youths who mentioned composition and the ingredients of disposable e-cigarettes, were concerned about the uncertainty of product ingredients and how they could affect their health.

> There's like about 40 or 50 chemicals that go into vapes that nobody in this room could name, all cheaply produced. So, see when you're inhaling it deep into your lungs it's obviously not going to be the best for you. (Male, never smoker, never vaper)

> I saw a thing on TikTok, Elfbars and Geek bars have got 50 unknown chemicals in them. (Male, never smoker, never vaper)

> Just see like the actual vapes instead of the disposables, they've all been tested. I don't think the disposables have been tested. (Male, never smoker, never vaper)

Several participants from different focus groups reported seeing people attempting to reuse the disposable e-cigarettes once they have been discarded.

> A lot of people will go and find them. It's weird. It's like people chuck them and other people go and find them and use them. (Male, never smoker, never vaper)

### Purchasing behaviours

Several participants commented on the low cost of disposable e-cigarettes.

> Like metal ones, I don't even know, I'm guessing around like 70 or £80, but then the disposable ones are like 6 to 12 or something like that. (Female, never smoker, never vaper)

With some participants commenting favourably on the relatively low cost of disposable e-cigarettes, suggesting that price could be a factor in why youths experiment with the products.

> They're cheap and cheerful. (Female, never smoker, current vaper)

> That's probably an attraction for young people because they're more affordable. (Female, never smoker, tried vaping)

Participants also described the ease of purchasing disposable e-cigarette products, particularly in corner shops.

> Like, I'm 16 and I buy Red Bull in there [corner shop] but I've got such a baby face. Like, I could walk into the shop and go, you're not 16. But if I was to buy a vape they would give me it, loads of folk underage buy them [disposable e-cigarettes] there. (Female, tried smoking, tried vaping)

Several participants discussed the ease of being able to purchase the products online as well.

> Some places, some websites online, you don't need to put your age or anything. I've seen a thing on TikTok. Like, they put them [disposable e-cigarettes] in the wee boxes and all that, or you could put them in secret packaging like behind the lashes. Like you can order it off their website and they'll hide it in the packaging, they put a few bits of sweeties on top of your vapes so your mum doesn't see it. (Female, tried smoking, current vaper)

### DISCUSSION

E-cigarettes have become increasingly popular and visible in public life and perceptions about e-cigarette users were tied to product characteristics, with tank models being associated with adults and disposable e-cigarettes associated with youths. The design of disposable e-cigarettes was a recurrent topic. Youth discussed the positives of the

compact design of the product as this allowed them to be discretely carried and hidden when in school. We found that youths commonly mistake the products for other everyday products, such as highlighters and tins of mints. This combined with the compact design of the products raises concerns about the way manufactures design the products and if this has been done intentionally to target a younger audience.

E-cigarette users believed that disposable e-cigarettes are less harmful than combustible cigarettes, while non-users reported concerns about the unknown chemical composition of disposable e-cigarettes. It is possible that if e-cigarette users perceive cigarettes as more harmful to their health they will be less likely to take up smoking and this may explain the potential displacement of cigarettes as reported in Williams *et al*.[2] This suggests it is important to track such changes in the population through longitudinal studies to detect and monitor youths perceptions, behaviours and assessment of risk in relation to e-cigarettes versus cigarettes. While e-cigarettes are considered less harmful than combustible cigarettes,[21 22] balanced policies are needed that motivate cigarette smokers to switch to e-cigarettes, yet prevent non-smokers or non-nicotine users from initiating, particularly youths.

The increased popularity of disposable e-cigarettes (such as PuffBar and ElfBar) has resulted in the generation of more single-use plastic waste. Both users and non-users were aware of the negative environmental impact of using disposable e-cigarettes. E-cigarettes remain subject to political and public health debates for various reasons, including the lack of evidence on their long-term health impact, and now there is a new topic in the scientific debate; disposable e-cigarettes are a rising environmental threat.[23 24] Thus, regulation should not only focus on the health effects of e-cigarette products, but may wish to consider their environmental impact.

Consistent with previous research,[25–29] our study found that participants particularly like the variety of disposable e-cigarette flavours and the variety of available flavours is one of the top reasons for experimentation with e-cigarettes among youths. Interestingly, in our study, participants discussed flavours predominately in relation to disposable e-cigarettes, often associating the colour of the product with its flavour. It was perceived from the youths in this study that disposable e-cigarettes are targeted to younger audiences. While rechargeable e-cigarettes (tank models) were perceived by our participants, as products for adults. Several studies[30–33] have recommended banning the sale of all flavoured e-cigarette products to help protect children and youth from the harms of vaping. However, some researchers argue that removing flavours will promote more combustible tobacco use and remove a product that facilitates smoking cessation[34 35] as research has shown that flavourings may help reduce the amount of cigarettes used by adult smokers in the short term.[8] In late 2022, China prohibited the domestic marketing and sales (including online) of flavoured disposable e-cigarettes, meaning e-cigarettes that have flavourings other than tobacco cannot be sold on the domestic market.[36 37] In addition, they have introduced regulations that all e-cigarette packaging must include warning labels stating that they are harmful to health and must not be used by school children.[36 37] Notably, flavoured disposable e-cigarettes can still be manufactured in China and shipped around the world, including to the UK. The Chinese government have stated that the devices must conform to the regulations of the importing country.[36 37]

More research is needed to determine the most effective means to counter the favourable/positive aspects of e-cigarettes to reduce youths' interest in product trial and use. In addition, more evidence is needed to determine what has contributed to the popularity of disposable e-cigarettes among youths, including, but not limited to, the role of marketing. These findings could inform future policies on e-cigarette prevention.

As with all research, our study has some limitations. First, and consistent with the qualitative design, the sample does not aim to be representative of UK youth, as our study focused on Scottish youths. However, we did have a diverse sample of both sexes. Second, the study's geographical remit has to be considered when interpreting the findings. The UK is considered an international leader in tobacco control policy. It is possible that participants' views may have been influenced by the UK's unique favourable policy approach to e-cigarettes and legal and sociocultural context, including low smoking prevalence. Third, the data were collected in different formats (online and face-to-face), and it is possible that this may have influenced participants' responses. Two of the online groups were conducted in a classroom with a teacher present, and during seven face-to-face groups, a teacher/youth worker was present in the room. It is possible that the presence of a teacher/youth worker may have influenced youth's willingness to answer questions and their responses. Finally, two of the groups were recruited through personal networks and this may have impacted on the youth's responses. Despite these limitations, our study results have implications for public health and policy. Results from our study highlighted that youths positively describe the relatively low cost of disposable e-cigarettes, suggesting that price could be a factor in why youths experiment with disposable e-cigarettes. Raising prices on combustible cigarettes and alcohol has consistently shown to be inversely related to use,[38 39] particularly among younger populations.[40 41] Therefore, policymakers could consider implementing measures to deter youth experimentation with disposable e-cigarettes, while not making the products inaccessible to vulnerable groups who may use them as a smoking cessation option. Our study suggests the growing need for policymakers to work together to develop and implement comprehensive policies to prevent initiation among youths and evaluate the safe recycling and disposal of disposable e-cigarettes. Our study suggests the growing need for policymakers to work together to develop and implement comprehensive

policies to prevent initiation among youths, such as through youth awareness programmes designed to prevent the start of e-cigarette use among youths which could include information on the effects of vaping the body, how to identify false marketing and how to resist peer pressure.[42] In addition, our research suggests policies are required to evaluate the safe recycling and disposal of disposable e-cigarettes (such as requiring manufacturers and retailers to instal collection points inside shops).

## CONCLUSION

We found that youths differentiated between disposable e-cigarettes and larger tank models, for which they had varying perceptions of product users. Our study highlights the need for additional research on e-cigarette subtypes to understand product perceptions more fully; and should be considered in future prevention and regulatory efforts. In addition, while many positive attributes of disposable e-cigarettes were reported, key negative attributes that may discourage use, such as unknown chemical composition and environmental impact, were also described. The findings from our study suggest the growing need for policymakers to work together to develop and implement policies to prevent uptake among youths.

**Contributors** MJS: guarantor, conceptualisation, data curation, investigation, methodology, validation, visualisation, writing—original draft preparation. AMM: conceptualisation, methodology, Writing—review and editing. AF: conceptualisation, methodology, writing—review and editing. SH: conceptualisation, methodology, validation, writing—review and editing.

**Funding** MJS, AF and AMM acknowledge funding from Cancer Research UK grant PPRCTAGPJT\100003. SH is funded by the Medical Research Council grant MC_UU_00022/1, the Chief Scientist Office of the Scottish Government Health Directorates grant SPHSU17, and Cancer Research UK grant PPRCTAGPJT\100003.

**Competing interests** None declared.

**Patient and public involvement** Patients and/or the public were not involved in the design, or conduct, or reporting or dissemination plans of this research.

**Patient consent for publication** Consent obtained directly from patient(s).

**Ethics approval** Ethical approval for the study was obtained from the University of Glasgow's Medical and Veterinary Life Sciences Ethics Committee (reference 200210034). Participants gave informed consent to participate in the study before taking part.

**Provenance and peer review** Not commissioned; externally peer reviewed.

**Data availability statement** All data relevant to the study are included in the article or uploaded as supplementary information.

**ORCID iDs**
Marissa J Smith http://orcid.org/0000-0002-5017-6085
Shona Hilton http://orcid.org/0000-0003-0633-8152

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
