## [Reviewer comments · BMJ Open]

ARTICLE DETAILS

TITLE (PROVISIONAL)	Youth's engagement and perceptions of disposable e-cigarettes: a UK focus group study
AUTHORS	Smith, Marissa; MacKintosh, Anne Marie; Ford, Allison; Hilton, Shona

VERSION 1 – REVIEW

REVIEWER	Kock, Loren University College London Research Department of Epidemiology and Public Health, Behavioural Science and Health
REVIEW RETURNED	19-Oct-2022

GENERAL COMMENTS	Overall a helpful and timely analysis of youth perceptions towards disposables. However I think the manuscript needs to be focussed and detailed in several areas. The introduction needs to set up why talking to youths is important to help understand and act on recent trends. The results could be discussed with more depth to consider what they mean in relation to the wider trends in vaping and smoking in the UK (reflecting on overall inhaled nicotine/impact on smoking uptake/displacement, regulation of flavored disposables rather than flavors in general if youth do not see these other products as appealing. Specific comments below! BACKGROUND 1. The first sentence of the background could be dropped as you mention differences in opinion with regards to harm reduction in Line 12 of page 4. However I feel you should try and outline what the issues are some more (i.e. growing consensus and evidence supporting e-cigs for smoking cessation among adults, while at same time there is the need to prevent uptake among youth). 2. From what I can gather, reference 12 does not cover the statement made about equivalence to cigarettes in line 44 of page 4. What are you stating as equivalent? The amount of nicotine delivered? This may be true but more direct evidence is needed. The paper cited (12) appears to compare pod devices to Juul. 3. The background is good overview of the current disposables situation in the UK, but more needs to be done to set up why discussing perceptions and engagement among youth is important, and what it can tell us. What can targeted interviews like this tell us more than generalizable population surveys? I think there is certainly value in it in terms of understanding the "why" behind the trends, and to help guide thinking around targeted policies that don't undermine the benefits for cessation among adults, but this is not
--

mentioned at all in the introduction.

METHODS

4. More detail needed on how you analyzed the data. Was there a particular method you employed? Framework analysis? Thematic analysis? Some more detail is needed as to this process beyond listing coding protocol and cross-checking. Were codes grouped under each pre-specified topic-guide theme? Did themes emerge outside of the topic guide? Did the researchers consider their role as researchers and interviewers and how this might have influenced their interpretation of responses?

5. Minor comment.. The statement on patient and public involvement might be a journal requirement but it strikes me as slightly odd! It may be that you didn't have the resource to do PPI (and that is fine!) but I think it is contestable that including participants in design, conduct, reporting or dissemination is considered "not appropriate". If in fact it is, why?

6. Why did you recruit individuals of different smoking and vaping status? Depending on the research question it is possible that different sub-groups might have nuanced perceptions which might get lost when grouping together responses across individuals with different smoking status. There may be a good reason for including the diverse sample, but it is not obvious to me as to why.

RESULTS

6. Page 6 line 48: The first sentence of the results seems like it is missing a word or two to connect the parts before and after the comma...

DISCUSSION

7. The first two sentences of the discussion are redundant and could be dropped. (lines 30-34)

8. Despite my comment above about the lack of clarity about recruiting youths of different smoking/vaping status, I can see the usefulness in the discussion where you mention e-cig users believing e-cigs are less harmful than tobacco cigarettes, and non-users reporting concerns. This is interesting and the implications could be drawn out some more. If e-cig users (who due to perhaps more common liability for other risk taking behaviors like smoking) perceive cigarettes as worse for their health then they will be less likely to take up smoking (which may help explain potential displacement of cigarettes you cite from the smoking toolkit study (no increases in inhaled nicotine - this of course may change).

9. Did you discuss issues related to marketing of disposables at youth through mediums such as tik Tok? Aside from the colors and sweeteners and availability, paid marketing on social media might be crucial and in need of regulation.

10. You may want to consider discussing what the recent ban on domestic marketing and sales of flavored disposables in China. Given that most disposables in the UK are manufactured in China this might have important implications for their availability. Notably,

	the Chinese government has not banned the export to other countries but has stated that the devices must conform to the regulations of the importing country. How regulation changes in the UK is therefore incredibly important in this context. 11. The finding about flavors being associated with disposables rather than larger tank devices is also very important in terms of regulation to help support adult smoking cessation (adults like flavors too) but in preventing youth uptake. More discussion needed on the implications of this perception. 12. Given what is at stake I think some discussion as to the implications of disposables on tobacco smoking among youth (reinforcing or displacing?). The current evidence from data you've cited (13) indicates that despite increases in disposables among youth, the overall prevalence of all inhaled nicotine has not increased. Other minor suggestions: Under "How this study might affect research, practice or policy" (and in the concluding paragraphs) the first half of the first bullet point is currently too vague to be helpful (of course it would be great if policymakers worked together to develop comprehensive policies to prevent initiation). Could you outline any specific policy areas that could be targeted based on your findings?
--	--

REVIEWER	Caponnetto, Pasquale Universita degli Studi di Catania Scuola di Facolta di Medicina PC has been affiliated with the CoEHAR since December 2019 in a pro bono role. He is co-author of a protocol paper supported by an Investigator-Initiated Study award program established by Philip Morris International in 2017.
REVIEW RETURNED	05-Dec-2022

GENERAL COMMENTS	More information about sample size and qualitative analysis are requested
---

VERSION 1 – AUTHOR RESPONSE

Reviewer: 1

Overall, a helpful and timely analysis of youth perceptions towards disposables. However, I think the manuscript needs to be focused and detailed in several areas. The introduction needs to set up why talking to youths is important to help understand and act on recent trends. The results could be discussed with more depth to consider what they mean in relation to the wider trends in vaping and smoking in the UK (reflecting on overall inhaled nicotine/impact on smoking uptake/displacement, regulation of flavoured disposables rather than flavours in general if youth do not see these other products as appealing. Specific comments below!

- We thank the reviewer for their helpful comments, please see below for a point-to-point response to each comment.

R1.1 The first sentence of the background could be dropped as you mention differences in opinion with regards to harm reduction in Line 12 of page 4. However, I feel you should try and outline what

the issues are some more (i.e., growing consensus and evidence supporting e-cigs for smoking cessation among adults, while at same time there is the need to prevent uptake among youth).

- We thank the reviewer for their comment. We have removed the first sentence of the background and have revised the background to outline the main issues. Please see the manuscript for our tracked changes.

R1.2 From what I can gather, reference 12 does not cover the statement made about equivalence to cigarettes in line 44 of page 4. What are you stating as equivalent? The amount of nicotine delivered? This may be true but more direct evidence is needed. The paper cited (12) appears to compare pod devices to Juul.

- We thank the reviewer for their comment. We agree that more direct evidence is needed and therefore have removed this section on nicotine delivery from the background section.

R1.3 The background is good overview of the current disposables situation in the UK, but more needs to be done to set up why discussing perceptions and engagement among youth is important, and what it can tell us. What can targeted interviews like this tell us more than generalizable population surveys? I think there is certainly value in it in terms of understanding the "why" behind the trends, and to help guide thinking around targeted policies that don't undermine the benefits for cessation among adults, but this is not mentioned at all in the introduction.

- We agree with the reviewer that more information is required on why we conducted discussion groups. We have revised the background as follows:

“User-generated and influencer marketing content on social media represents a key influence on young people’s understandings of products. It is essential to monitor the content that young people access related to e-cigarettes and through focus groups with youths so we can understand how young people relate to that content, why e-cigarettes might appeal to youths, and why they need protected, which would not be feasible with population surveys.”

R1.4 More detail is needed on how you analysed the data. Was there a particular method you employed? Framework analysis? Thematic analysis? Some more detail is needed as to this process beyond listing coding protocol and cross-checking. Were codes grouped under each pre-specified topic-guide theme? Did themes emerge outside of the topic guide? Did the researchers consider their role as researchers and interviewers and how this might have influenced their interpretation of responses?

- We thank the reviewer for their comment. We have substantially revised the methods section of the manuscript to include more detail. Please see the tracked manuscript for our changes.

R1.5 The statement on patient and public involvement might be a journal requirement but it strikes me as slightly odd! It may be that you didn't have the resource to do PPI (and that is fine!) but I think it is contestable that including participants in design, conduct, reporting or dissemination is considered "not appropriate". If in fact it is, why?

- We thank the reviewer for their comment. We apologise for the use of the terminology in this section. We used a generic statement and upon reflection believe it is not appropriate. We have revised the

manuscript as follows:

“Patient and public involvement
None.”

R1.6 Why did you recruit individuals of different smoking and vaping status? Depending on the research question it is possible that different sub-groups might have nuanced perceptions which might get lost when grouping together responses across individuals with different smoking status. There may be a good reason for including the diverse sample, but it is not obvious to me as to why.

- We thank the reviewer for their comment. We recruited participants with different smoking and vaping statuses as we believed they may have different perspectives on product types, messages and presentations. We were not able to say this for certain during recruitment but having analysed the results we found that e-cigarette users perceived e-cigarettes as less harmful than tobacco cigarettes, while non-users reported concerns. In response to R1.9, we have revised the discussion to address the implications of this. Please see our response to R1.9.

R1.7 Page 6 line 48: The first sentence of the results seems like it is missing a word or two to connect the parts before and after the comma...

- We thank the reviewer for highlighting this error. We have revised the manuscript as follows:

“Eighty-two youths aged 11–16 years participated (47 females (57%) and 35 males (43%)). This sample represented a wide diversity in sociodemographic characteristics and smoking-related behaviours. The age distribution within the sample was skewed slightly towards 14–15-year-olds, with 14-year-olds making up the largest subgroup (n= 24).”

R1.8 The first two sentences of the discussion are redundant and could be dropped. (lines 30-34)

- We agree with the reviewer and have removed the first two sentences of the discussion.

R1.9 Despite my comment above about the lack of clarity about recruiting youths of different smoking/vaping status, I can see the usefulness in the discussion where you mention e-cig users believing e-cigs are less harmful than tobacco cigarettes, and non-users reporting concerns. This is interesting and the implications could be drawn out some more. If e-cig users (who due to perhaps more common liability for other risk taking behaviours like smoking) perceive cigarettes as worse for their health then they will be less likely to take up smoking (which may help explain potential displacement of cigarettes you cite from the smoking toolkit study (no increases in inhaled nicotine - this of course may change).

- We thank the reviewer for this comment and have revised the discussion, please see below for our changes:

“It is possible that if e-cigarette users perceive cigarettes as more harmful to their health they will be less likely to take up smoking and this may explain the potential displacement of cigarettes as reported in [1]. This suggests it is important to track such changes in the population through longitudinal studies to detect and monitor youths perceptions, behaviours and assessment of risk in relation to e-cigarettes verse cigarettes.”

R1.10 Did you discuss issues related to marketing of disposables at youth through mediums such as tik Tok? Aside from the colours and sweeteners and availability, paid marketing on social media might be crucial and in need of regulation.

- We thank the reviewer for this comment. We discussed marketing and advertising of e-cigarettes on social media, and this was one of the two main topics to emerge from the focus groups, the other being disposable e-cigarettes. We have submitted a paper which focuses on the advertising and marketing of e-cigarettes on social media and it does mention Tik Tok. We believed that the data relating to advertising was best suited to an individual paper.

R1.11 You may want to consider discussing what the recent ban on domestic marketing and sales of flavoured disposables in China. Given that most disposables in the UK are manufactured in China this might have important implications for their availability. Notably, the Chinese government has not banned the export to other countries but has stated that the devices must conform to the regulations of the importing country. How regulation changes in the UK is therefore incredibly important in this context.

- We agree with the reviewer that it would be pertinent to discuss the recent ban of marketing and sale of flavoured disposable e-cigarettes in China. We have revised the manuscript as follows:

“In late 2022, China prohibited the domestic marketing and sales (including online) of flavoured disposables e-cigarettes, meaning e-cigarette that have flavourings other than tobacco cannot be sold on the domestic market [2, 3]. In addition, they have introduced regulations that all e-cigarette packaging must include warning labels stating that they are harmful to health and must not be used by schoolchildren [2, 3]. Notably, flavoured disposable e-cigarettes can still be manufactured in China and shipped around the world, including to the UK. The Chinese government have stated that the devices must conform to the regulations of the importing country [2, 3].”

R1.12 The finding about flavours being associated with disposables rather than larger tank devices is also very important in terms of regulation to help support adult smoking cessation (adults like flavours too) but in preventing youth uptake. More discussion needed on the implications of this perception.

- We agree with the reviewer that it would be pertinent to discuss implications of flavourings. We have revised the manuscript as follows:

“Several studies [4-7] have recommended banning the sale of all flavoured e-cigarette products to help protect children and youth from the harms of vaping. However, some researchers argue that removing flavours will promote more combustible tobacco use and remove a product that facilitates smoking cessation [8, 9] as research has shown that flavourings may help reduce the amount of cigarettes used by adult smokers in the short term [10].”

R1.13 Given what is at stake I think some discussion as to the implications of disposables on tobacco smoking among youth (reinforcing or displacing?). The current evidence from data you've cited (13) indicates that despite increases in disposables among youth, the overall prevalence of all inhaled nicotine has not increased.

- We agree with the reviewer that more detail is required in relation to disposable e-cigarettes and

tobacco smoking among youth.

"It is possible that if e-cigarette users perceive cigarettes as more harmful to their health they will be less likely to take up smoking and this may explain the potential displacement of cigarettes as reported in [1]. This suggests it is important to track such changes in the population through longitudinal studies to detect and monitor youths' perceptions, behaviours and assessment of risk in relation to e-cigarettes versus cigarettes."

R1.14 Under "How this study might affect research, practice or policy" (and in the concluding paragraphs) the first half of the first bullet point is currently too vague to be helpful (of course it would be great if policymakers worked together to develop comprehensive policies to prevent initiation). Could you outline any specific policy areas that could be targeted based on your findings?

• We thank the reviewer for their comment. The section on "How this study might affect research, practice or policy" was a required section of another journal. While it is not required for BMJ Open we think it is still important to discuss implications for practice or policy. We have revised the discussion section to include details from this original section and as suggested we have revised our original statement.

"Our study suggests the growing need for policymakers to work together to develop and implement comprehensive policies to prevent initiation among youths, such as through youth awareness programs designed to prevent the start of e-cigarette use among youths which could include information on the effects of vaping the body, how to identify false marketing, and how to resist peer pressure [11]. In addition, our research suggests policies are required to evaluate the safe recycling and disposal of disposable e-cigarettes (such as requiring manufacturers and retailers to install collection points inside shops).

Reviewer: 2

R2.1 More information about sample size and qualitative analysis are requested.

• We thank the reviewer for their comment. We have substantially revised the methods section of the manuscript to include more detail on sample size and analysis. Please see the tracked manuscript for our changes.

References

1. Williams P, Cheeseman H, Arnott D, Bunce L, Hopkinson NS, Lavery AA. Use of tobacco and e-cigarettes among youth in Great Britain in 2022: analysis of a cross sectional survey. 2022.
2. Das S, Ungood-Thomas J, Y L. China bans fruity vapes – but not their export to the UK. The Observer. 2022.
3. Coy K. The Beijinger. 2022 27 September 2022. [cited 2023]. Available from: <https://www.thebeijinger.com/blog/2022/09/27/delayed-ban-sale-non-tobacco-flavored-vapes-set-go-ahead-oct-1>.
4. Chadi N, Vyver E, Bélanger RE. Protecting children and adolescents against the risks of vaping. 2021;26(6):358-65.
5. Kingsley M, Setodji CM, Pane JD, Shadel WG, Song G, Robertson J, et al. Short-Term Impact of a Flavored Tobacco Restriction: Changes in Youth Tobacco Use in a Massachusetts Community. 2019;57(6):741-8.
6. Walley SC, Jenssen BP. Electronic Nicotine Delivery Systems. 2015;136(5):1018-26.
7. Campaign for Tobacco-Free Kids. The flavor trap: how tobacco companies are luring kids with

- candy-flavored e-cigarettes and cigars. *American Academy of Pediatrics*; 2017.
8. Polosa R, Caponnetto P, Maglia M, Morjaria JB, Russo C. Success rates with nicotine personal vaporizers: a prospective 6-month pilot study of smokers not intending to quit. 2014;14(1):1159.
 9. Adriaens K, Van Gucht D, Declerck P, Baeyens F. Effectiveness of the electronic cigarette: An eight-week Flemish study with six-month follow-up on smoking reduction, craving and experienced benefits and complaints. *Int J Environ Res Public Health*. 2014;11(11):11220-48.
 10. Hajek P, Phillips-Waller A, Przulj D, Pesola F, Myers Smith K, Bisal N, et al. A Randomized Trial of E-Cigarettes versus Nicotine-Replacement Therapy. *NEJM*. 2019;380(7):629-37.
 11. Kelder SH, Mantey DS, Van Dusen D, Vaughn T, Bianco M, Springer AE. Dissemination of CATCH My Breath, a middle school E-Cigarette prevention program. 2021;113:106698.